# Beneficial Effects of Marine Algae-Derived Carbohydrates for Skin Health

**DOI:** 10.3390/md16110459

**Published:** 2018-11-21

**Authors:** Ji Hye Kim, Jae-Eun Lee, Kyoung Heon Kim, Nam Joo Kang

**Affiliations:** 1School of Food Science and Biotechnology, Kyungpook National University, Daegu 41566, Korea; friend8201@naver.com (J.H.K.); lju1033@naver.com (J.-E.L.); 2Korean Medicine Application Center, Korea Institute of Oriental Medicine, Daegu 41566, Korea; 3Department of Biotechnology, Graduate School, Korea University, Seoul 02841, Korea

**Keywords:** marine algae, carbohydrates, oligosaccharides, monosaccharides, skin health, cosmeceuticals

## Abstract

Marine algae are considered to be an abundant sources of bioactive compounds with cosmeceutical potential. Recently, a great deal of interest has focused on the health-promoting effects of marine bioactive compounds. Carbohydrates are the major and abundant constituent of marine algae and have been utilized in cosmetic formulations, as moisturizing and thickening agents for example. In addition, marine carbohydrates have been suggested as promising bioactive biomaterials for their various properties beneficial to skin, including antioxidant, anti-melanogenic and skin anti-aging properties. Therefore, marine algae carbohydrates have potential skin health benefits for value-added cosmeceutical applications. The present review focuses on the various biological capacities and potential skin health benefits of bioactive marine carbohydrates.

## 1. Introduction

Cosmeceuticals can be defined as cosmetic products with biologically active ingredients purporting to exert pharmaceutical effects on the skin. Recently, great interest has been shown by consumers in novel bioactive compounds from natural sources, instead of synthetic ingredients, thanks to their perceived beneficial effects [1]. Therefore, there are numerous efforts to develop biologically active ingredients from natural organisms [2]. Most studies have been based on terrestrial sources; however, it has been shown that natural compounds isolated from marine sources show higher biological activity than those isolated from terrestrial sources, and as a result, there is a lot of interest in the studies of ingredients using natural marine sources [3,4]. In particular, oceans account for about 70% of the earth’s surface and their biodiversity makes them an excellent reservoir of sources for natural products [5]. Among various natural organisms, marine algae, which grow much faster than terrestrial plants, are considered to be abundant and essential sources of numerous constituents beneficial for human skin health [2,6].

Algae are photosynthetic organisms with a complex and controversial taxonomy [7]. To date, more than 20,000 species of algae have been identified, and there are two kinds of algae depending on size [6]. Macroalgae (seaweeds) are defined as multicellular marine plants that live in coastal areas and have simpler structures than terrestrial plants [6]. Marine macroalgae are classified into three species according to their pigments: *Phaeophyceae* (brown macroalgae, *Chromophyta*), *Chlorophyta* (green macroalgae) and *Rhodophyta* (red macroalgae) [6,8]. In contrast, microalgae are small unicellular or simple multicellular species and are found in various environments [6,7].

Marine algae are composed of various substances including carbohydrates, lipids, proteins, amino acids, minerals and flavonoids [9]. Among the various ingredients, carbohydrates are the most abundant constituents of marine algae [1,10,11]. Based on degrees of polymerization (DPs), carbohydrates, also called saccharides, exist in marine algae as various forms of monosaccharides, disaccharides, oligosaccharides and polysaccharides [1]. Marine carbohydrates have been utilized in cosmeceutical industries due to their chemical and physical properties [12,13]. Fucoidans/alginate from brown algae, ulvans from green algae and carrageenans/agar from red algae are used as gelling, thickening and stabilizing agents [2,6,12,14]. In addition, accumulating reports suggest that marine carbohydrates have been proven to exhibit potential benefits for skin [2,12]. The biological activities of marine carbohydrates are known to be linked with their structure as determined by DPs ormolecular weights, the presence of sulfate groups and types of sugars [15]. Therefore, in this review, we discuss the skin health cosmetic effects of carbohydrates extracted from marine algae, which are considered to be sources of excellent carbohydrates.

## 2. Bioactive Effects and Potential Health Benefits of Marine Algae

### 2.1. Biological Activities of Marine Algal Extracts

Table 1 shows the beneficial effects of marine algal extracts, including macroalgae and microalgae, for skin health.

#### 2.1.1. Macroalgal Extracts

Cha et al. screened 43 indigenous marine algae for new skin-whitening agents [16]. The aqueous extracts from brown algae *Endarachne binghamiae*, *Sargassum silquastrum*, *Ecklonia cava* and red algae *Schizymenia dubyi* exhibited potent mushroom tyrosinase (TYR) inhibitory activity. Both *E. cava* and *S. silquastrum* reduced cellular melanin synthesis and TYR activity in a murine cell model and zebrafish model at non-toxic concentrations. Heo et al. recently screened 21 species of marine algae for effects on melanogenesis using mushroom TYR activity [17]. Extracts of *Ishige okamurae Yendo* inhibited mushroom TYR activity and melanin synthesis in murine melanoma B16F10 cells.

According to Quah et al., ethanol or hexane extract of brown algae, including *Sargassum polycystum* and *Padina tenuis,* significantly reduced mushroom TYR activity and melanin content in human epidermal melanocytes (HEMs) [18]. Topical application with ethanol or hexane extract of *S. polycystum* attenuated melanin production in guinea pigs in dermal irritation tests and de-pigmentation assessments. Hexane extract of *S. polycystum* was the most potent without toxicity for in vitro and in vivo models.

Murugan et al. reported the antioxidant activity of extracts of brown, green and red marine algae. In vitro 2,2-diphenyl-1-picrylhydrazyl (DPPH) radical scavenging assay and ferrous ion chelation were performed with methanol (M), chloroform (C), ethyl acetate (EAc), and aqueous (A) extracts of *Sargassum wightii* (brown algae), *Padina gymnospora* (brown algae), *Caulerpa peltata* (green algae) and *Gelidiella acerosa* (red algae) [19]. Non-polar C and EAc extracts showed higher DPPH radical-scavenging. However, A extracts (polar extracts) showed higher ferrous ion chelation. These results suggest that the antioxidant activity of marine algal extracts may relieve skin aging and skin inflammation processes that are affected by oxidative stress [31].

In 2002, Fujimura’s group found that topical application of brown algae *Fucus vesiculosus* (Bladder wrack) aqueous extracts improved the thickness and elasticity of human cheek skin [20]. These results suggest that the *F. vesiculosus* extract possesses anti-aging activities and may be useful for a variety of cosmetics [20].

Previous study has shown the photoprotective effects of cosmetic formulations containing ultraviolet (UV) filters, vitamins, *Ginkgo biloba* and red algae *Porphyra umbilicalis* extracts for in vitro and in vivo models [28]. Topical formulations including F (sunscreen formulation containing only UV filters), FA (sunscreen formulation with red algae extract) and FVGA (sunscreen formulation with red algae extract, *G. biloba* and vitamins A, C and E) were applied on hairless mice. Extracts from the red algae *P. umbilicalis* could be considered effective ingredients for use in sunscreen formulations. The combination of vitamins A, E, C and *G. biloba* along with red algae extracts can significantly improve the performance of the sunscreens, preventing UV-induced DNA damage and inflammation. Al-Bader et al. reported the potential of skin anti-aging cosmetic ingredients containing red algae *Furcellaria lumbricalis* (black carrageen) and brown algae *Fucus vesiculosus* [29]. A mixture of *F. vesiculosus* and *F. lumbricalis* extracts induced expression of type 1 pro-collagen in aged human dermal fibroblasts (HDFs). Another clinical study demonstrated the skin anti-aging effects of *Spirulina maxima* (blue algae), *Ulva lactuca* (green algae) and *Lola implexa* (green algae) with other compounds [30]. Marine algal mixtures enhanced the skin hydrating and skin firming effects on human skin, suggesting the utilization of marine algae in cosmeceuticals.

#### 2.1.2. Microalgal Extracts

Skin anti-aging and skin barrier functions of microalgae extracts were assessed in vitro and in vivo [21]. Green-blue microalgae, *Blue Lagoon coccoid Filamentous,* were extracted with phosphate-buffered saline (PBS) without magnesium (Mg) and calcium (Ca). In human epidermal keratinocytes (HEKs), green-blue microalgae extracts increased the expression genes of the transcriptional level of involucrin (INV), loricrin (LOR), transglutaminase-1 (TGM-1) and filaggrin (FLG) which are major markers for skin barrier function [32]. UV radiation upregulates collagen degradation through the increase of matrix metalloproteinase-1 (MMP-1) expression in HDFs. *Blue Lagoon* extracts suppressed MMP-1 upregulation and type 1 pro-collagen downregulation stimulated by ultraviolet A (UVA). Topical treatment with *Blue Lagoon* extracts (0.25% and 2.5%) consistently reduced levels of transepidermal water loss (TEWL) in human skin. Collectively, *Blue Lagoon* extracts improved skin barrier function and showed a capacity to prevent premature skin aging.

Buono et al. demonstrated that aqueous extracts of *Botryococcus braunii* exhibited antioxidant, skin anti-aging and anti-inflammatory capacities in various cell-based models [22]. Skin aging is driven by oxidative stress in skin caused by intrinsic and extrinsic factors [31]. Oxygen radical absorbance capacity (ORAC) assay and COMET assay showed that intracellular reactive oxygen species (ROS) levels and DNA damage were decreased by *B. braunii* extracts in NIH3T3 mouse embryo fibroblasts. Decreased levels of aquaporin-3 (AQP3) and FLG, INV and pro-collagen were observed in aged skin [33,34]. *B. braunii* extract treatment increased expression of AQP3, FLG, INV and type 1 and 3 pro-collagen in HaCaT cells, indicating potential skin anti-aging activity. Antioxidant activity is also closely related to anti-inflammatory processes [31]. During inflammation, some pro-inflammatory cytokines and endotoxins induce the expression of an inducible nitric oxide synthase (iNOS), leading to the generation of nitric oxide (NO) in macrophages. Data revealed that *B. braunii* extracts significantly reduced lipopolysaccharide (LPS)-induced iNOS expression and NO production in murine macrophage RAW 264.7 cells. These results asserted that *B. braunii* water extract had been proved to exert biological activities consistent with skin health maintenance.

Several studies described diverse beneficial effects of aqueous extracts of green microalgae *Chlorella* for skin health. Kang et al. reported *Chlorella vulgaris* attenuates *Dermatophagoides Farinae* (DFE)-induced atopic dermatitis (AD) in NC/Nga mice [23]. Hidalgo-Lucas et al. reported that oral and topical administration of *Chlorella sorokiniana* (ROQUETTE *Chlorella* sp.) extracts improved skin inflammation induced by 12-O-tetradecanoylphorbol-13-acetate (TPA) in hairless Skh-1 mice [24]. A previous study assessed the chemopreventive potential of *C. vulgaris* against murine skin papillomagenesis [25]. Topical application of *C. vulgaris* (500 mg/kg b.w./day) significantly attenuated 12-dimethylbenz [a] anthracene (DMBA)-induced tumor size and number by upregulating the sulfhydryl (-SH) and glutathlone S-transferase (GST) levels in skin tissues. The results indicated that marine algae could be utilized as preventive and therapeutic agents for various inflammatory skin diseases.

Recently, spring water extracts of *Schizochytrium* (ROQUETTE *Schizochytrium* sp.) were reported to exert skin anti-inflammatory potential in vivo [26]. TPA-induced skin inflammation was significantly attenuated by oral administration (125, 250 and 500 mg/kg) and cutaneous application (2.5%, 5% and 10%) with *Schizochytrium* extracts in Skh-1 hairless mice. However, further studies are required to examine the active ingredients and to understand details of the molecular mechanism(s) and direct target(s).

Kim et al. reported the modulatory ability of 80% methanol extract of *Porphyra yezoensis* (laver) on ultraviolet B (UVB)-induced cell death in immortalized human keratinocyte, HaCaT cells [27]. The *P.yezoensis* extract can modulate cell viability and apoptosis of UVB-exposed cells via activating c-Jun N-terminal kinase (JNK) and extracellular signal–regulated kinase (ERK) signaling pathways, in which the modulation of redox status and content of glutathione by the extract. The results indicate that *P.yezoensis* extract can protect skin cells from UVB damage, contributing to improved skin health.

### 2.2. Biological Activities of Polysaccharides from Marine Algae

Marine algae are abundant in polysaccharides, such as fucoidans in brown algae, ulvans in green algae and carrageenans in red algae [35]. The beneficial effects on skin of polysaccharides from marine algae are summarized in Table 2 along with the species, biological function and mechanism of action.

#### 2.2.1. Fucoidans

Fucoidans are major sulfated polysaccharides (SPs) found in the cell wall of some brown algae [10]. Numerous studies have reported the benefits of fucoidans for diverse skin disorders including pigmentation, skin aging, atopic dermatitis and skin carcinogenesis.

##### Anti-Melanogenic Activity

Song et al. reported that fucoidan reduced melanin content by activating the ERK pathway in Mel-Ab Cells [36]. While fucoidan treatment did not directly decrease TYR activity, it downregulated the microphthalmia-associated transcription factor (MITF) and TYR protein expression.

##### Antioxidant Activity

In vitro antioxidant capacities of fucoidan from *Sargassum tenerrimum* were analyzed with DPPH, superoxide radical scavenging and total antioxidant assays [40]. The antioxidant activity of SPs depends on their structural properties such as the level or distribution of sulfate groups, MW, sugar composition, and stereochemistry [35]. It has been consistently documented that fucoidan from brown algae *Laminaria japonica* possesses high superoxide radical and hydroxyl radical scavenging assays according to sulfate content [38,39]. Fucoidans from *Fucus vesiculosus* exhibited considerable ferric reducing antioxidant power (FRAP) [37] and superoxide radical scavenging property [68].

##### Skin Anti-Aging Activity

A study conducted by Moon et al. reported that fucoidan from *Costaria costata* showed skin anti-aging activity in human foreskin fibroblast HS68 cells [41,42] and HaCaT cells [43].

Fucoidan suppressed mRNA and protein expression of MMP-1 upregulation and type 1 pro-collagen downregulation stimulated by UVB via inactivation of ERK and JNK. Additionally, fucoidan from *Mekabu* inhibited Interleukin-1β (IL-1β)-induced secretion of MMP-9, -3 and degradation of tissue inhibitor of metalloproteinases inhibitor 1 (TIMP-1) in HDFs [45]. In addition, positive correlations reported for UVB-induced edema, thickness of the prickle cell layer, MMP-1 activation and interferon (IFN)-γ were attenuated by fucoidan treatment on the skin of mice [44]. Senni et al. demonstrated that fucoidan (16 kDa) from *Ascophyllum nodosum,* using acidic hydrolysis, exhibited skin anti-aging potential in human skin via preventing elastic fiber degradation and leukocyte elastase activity [45]. These results indicate that fucoidans present skin anti-aging potential with varied mechanisms of action.

##### Anti-Atopic Dermatitis Activity

Fucoidan from *Laminaria cichorioides* alleviated 2,4-dinitrochlorobenzene (DNCB)-induced AD in vitro and in vivo [46]. AD-associated chemokines including thymus- and activation-regulated chemokine (TARC), macrophage-derived chemokine (MDC) and regulated upon activation, normal T-cell expressed and secreted chemokine (RANTES), were inhibited by fucoidan treatment in human keratinocytes. Another study reported anti-atopic dermatitis effects ex vivo whereby fucoidan inhibited IgE production in peripheral blood mononuclear cells (PBMC) from patients with AD, as well as immunoglobulin germline transcripts of B cells and the IgE-secreting cell count [47]. Thus, fucoidan could contribute to the development of preventive and therapeutic agents for inflammatory diseases such as AD.

##### Moisturizing Activity

Previously, *Saccharina japonica* extracts from brown algae showed a profound moisture retention ability, greater than that of other kinds of algae [6]. In particular, *S. japonica* polysaccharides were identified as a better humectant than hyaluronic acid (HA, or hyaluronan), which has the ability to retain a large amount of water [48], followed by red macroalgae extracts. Other extracts from green algae showed lower water retention capacity than HA. Therefore, SPs from marine algae, especially fucoidan, have potential as humectants to protect against skin dehydration.

##### Anti-Skin Cancer Activity

The chemopreventive activity and the underlying molecular mechanisms of fucoidan from *Laminaria cichorioides* was elucidated by Lee et al. in 2008. Treatments with water-soluble fucoidan from *L. cichorioides* up to 100 μg/mL were not cytotoxice in JB6Cl41 mouse epidermal cells. Fucoidan inhibited the epidermal growth factor (EGF), or TPA-induced neoplastic cell transformation, through preventing the binding of EGF to its cell surface receptor (EGFR) [49]. This evidence suggests an anti-skin carcinogenic molecular mechanism action of fucoidan with potential application for chemopreventive agents.

#### 2.2.2. Laminaran

Laminaran (also known as laminarin) is one of the major non-SPs found in brown algae. The biological activities of fucoidans have been well-studied, while those of laminaran have been poorly understood to date. Laminaran from the brown algae *Saccharina longicruris* has been reported to show skin anti-aging induced by UVA/UVB in an in vivo model [50]. The Kumming mouse is an experimental animal model reflecting age-related decline characteristics of female fertility in humans [69]. UV irradiation facilitates the process of extrinsic aging as well as intrinsic aging. Intraperitoneal (IP) injection of laminaran (1 or 5 mg/kg) attenuated UVA/UVB-induced skin dermal thickness by downregulating MMP-1 and upregulating TIMP-1 and hydroxyproline (Hyp) content. Ayoub et al. demonstrated that laminaran from *Saccharina longicruris* prevented matrix deposition [51]. Considering these results, laminaran may help prevent the progression of skin aging.

#### 2.2.3. Ulvans

Ulvans are sulfated heteropolysaccharides extracted from the cell wall of green algae *Ulva pertusa* [52]. Ulvans are water-soluble sulfated polysaccharides and their main constituents are rhamnose, xylose, glucose, uronic acid and sulfate. It has also been identified that glucuronic acid and rhamnose occur mainly in the form of the aldobiouronic acid, 4-*O*-β-D-glucuronosyl-l-rhamnose [70]. Due to the high recalcitrance of ulvans, related to their complex chemical structure, their biological functions have been less exploited.

Radical scavenging assay revealed that the antioxidant, reducing activity and ferrous ion chelating ability of ulvans were proportionate to sulfate content [52]. High sulfate content showed more profound antioxidant properties [52]. A follow-up study reported that low molecular weight (LMW) and high sulfate content derivatives of ulvans showed enhanced antioxidant activities [35]. In addition, the antioxidant activity of acetylated and benzoylated ulvans was stronger than that of natural ulvan [53]. Recently, SPs including crude ulvans (57 kDa) and LMW ulvan (4 kDa) were isolated from *Ulva* sp. and their skin anti-aging activities were evaluated [54]. HA production was significantly upregulated by SPs from *Ulva* sp. in HDFs. Crude ulvans (57 kDa) showed stronger stimulatory activity of HA production than LMW ulvan (4 kDa). These findings revealed the biological activities of ulvans and may account for the development of ingredients beneficial to skin from marine algae.

#### 2.2.4. Porphyran

Red algae *Porphyra* is an edible seaweed well-known as laver, gim (Korean) or nori (Japanese). Porphyra is mainly composed of porphyran, which is the sulfated polysaccharide comprising the hot-water soluble portion of the cell wall [58]. Porphyran is related to agarose in that it contains disaccharide units consisting of 3-linked β-d-galactosyl residues alternating with 4-linked 3,6-anhydro-α-L-galactose, but differs in that some residues occur as 6-sulfate [57].

##### Antioxidant Activity

Porphyran has been reported to scavenge oxidative radicals in vitro [55], and to increase antioxidant enzyme activity and antioxidant capacity in aging mice [56,57].

Porphyran from *Porphyra* sp. aqueous extract showed significant ferrous ion chelating capacity and reducing power [55]. In addition, DPPH radicals and superoxide radicals were dose-dependently quenched by porphyran treatment. Zhao et al. [58] found that the porphyrans from *Porphyra haitanensis* with different MWs showed different antioxidant activities. Assays including DPPH radical and reducing power indicated that porphyrans with lower MW exhibited higher antioxidant activities. According to a follow-up study, LMW porphyran and its different derivatives determined the relationship between antioxidant activity and chemical modifications [59]. Sulfated (SD), acetylated (AD), phosphorylated (PD) and benzoylated (BD) derivatives of porphyran from *P. haitanensis* showed higher antioxidant activities in vitro than those of LMW porphyran. Among the diverse derivatives, BD exerted the best antioxidant activities in DPPH radical, hydroxyl radical and superoxide scavenging assays. These results also support the conclusion that the antioxidant activity of polysaccharide is closely related to several structural elements such as MW, degree of substitution (DS) and functional groups [59].

In vivo antioxidant activity of porphyran fraction F1 [57] and F2 [56] derived from *P. haitanensis* has been assessed in aging mice (Kumming mouse) [35,69]. Malondialdehyde (MDA) is a main marker of endogenous lipid peroxidation. With aging, the organs showed significantly increased levels of MDA indicating that peroxidative damage increases with the aging process [56,57]. IP administration of porphyran fractions F1 (50, 100 and 200 mg/kg) and F2 (100, 200 and 400 mg/kg) significantly decreased the MDA level in aging mice, indicating a prevention effect of lipid peroxidation. Superoxide dismutase (SOD) is an intracellular antioxidant enzyme that protects against oxidative processes initiated by the superoxide anion [57]. Glutathione peroxidase (GSH-Px) is an enzymatic antioxidant defense system to protect against oxidative damage, while total antioxidant capacity (TAOC) reflects the capacity of the non-enzymatic antioxidant defense system [57]. Porphyran fractions F1 and F2 both increased the TAOC and upregulated activity of SOD and GSH-Px in Kumming aging mice suggesting their significant in vivo antioxidant activity [56,57].

##### Skin Anti-Inflammatory Activity

Porphyran from *Porphyra yezoensis* showed the anti-inflammation activity in LPS-stimulated macrophages [60]. Porphyran suppressed LPS-induced NO production and iNOS level by blocking nuclear factor kappa B (NF-κB) activation in RAW264.7 cells. Porphyran reduced LPS-induced NF-κB activation by inhibiting nuclear translocation of p65, phosphorylation and degradation of inhibitor of kappa B (IκB)-α in RAW264.7 cells. Meanwhile, porphyran showed a moderate inhibitory effect on LPS-induced tumor necrosis factor (TNF)-α production in RAW264.7 cells. These results suggest that porphyran blocked LPS-induced NO production via inactivation of NF-κB in murine macrophage cells.

#### 2.2.5. Carrageenan

Carrageenan from red algae is linear SP composed of 3,6-anhydro-D-galactose (D-AHG) and d-galactose. Carrageenan has been utilized in cosmetic products as a stabilizer, emulsifier and moisturizer due to its chemical and physical properties. In addition, carrageenan is known to exhibit various beneficial effects on skin health as summarized in Table 2.

##### Anti-Melanogenic Activity

Carrageenan from red microalgae *Porphyridium*, has been reported as being a macrophage toxic substance [62]. The injection of carrageenan effectively degraded and eliminated dermal melanosomes/melanin from the dermis of guinea pigs indicating the skin-whitening potential of carrageenan.

##### Antioxidant Activity

Thevanayagam et al. assessed the photoprotective and antioxidative activities of various isoforms of carrageenan in HaCaT cells [63]. The types of carrageenan are iota 2 [ι (ІІ)] iota 5 [ι (V)] from *Eucheuma spinosum*, and lambda (λ) and kappa (κ) type ІІІ from *Eucheuma cottonii*. Commonly, all types of carrageenan can scavenge free radicals, however, in vitro antioxidant capability did not correlate with the amount of sulfur moieties in the different isomers. Although κ-carrageenan contained the least sulfate content compared to ι- and λ-carrageenan, κ-carrageenan exhibited the highest radical scavenging activity. The DPPH reducing capability of carrageenan followed the order: λ < ι < κ. This evidence indicates that the increase in the oxidative property with irradiation dose can be attributed mainly to the depolymerization of the carrageenan with a corresponding increase in reducing sugar. In addition, the presence of the hydrophobic 3,6-anhydrogalatose could affect the antioxidant activity of carrageenan.

Other studies investigated the antioxidant capacity of κ-carrageenan, κ-carrageenan oligosaccharides (κ-COSs) and their chemically modified derivatives including oversulfated (SD, 0.8 kDa), lowly acetylated (LAD, 1.2 kDa), highly acetylated (HAD, 1.4 kDa) and phosphorylated (PD, 1.1 kDa) [64,65,66]. An in vitro antioxidant activity assay was performed reducing power, iron ion chelation, and total antioxidant activity. Generally, chemical modification of COSs can enhance their antioxidant activity in vitro as follows: PD > SD > LAD > HAD [66]. In this study, sulfate contents seemed to be related to antioxidant activity. Taken together, these investigations indicate that the antioxidant properties of carrageenans have are closely related to sulfate content structure as well as with the type of sugar unit and DPs according to MW.

##### Photoprotective Activity

Ren et al. reported the anti-oxidative and photoprotective effects of a complex of κ-COSs and collagen peptide (CP) in HaCaT cells and mouse embryonic fibroblasts (MEFs) [67]. A complex of κ-COSs and CP (100 μg/mL) could significantly attenuate UV-induced cell death and apoptosis in HaCaT and MEF through reduction of the intracellular ROS level. A complex of κ-COSs and CP mostly inhibited the UV-induced decrease of type 1 pro-collagen and increase in MMP-1 by suppressing the mitogen-activated protein kinases (MAPKs) signaling pathway. Collectively, a complex of κ-COSs and CP may have photoprotective potential against skin aging.

### 2.3. Biological Activities of Monosaccharides and Oligosaccharides from Red Algae

Agar is the major polysaccharide of red macroalgae. Agar is easily hydrolyzed into oligosaccharides by various chemical and enzymatic methods [71]. Depending on the hydrolysis method, oligosaccharides with different DPs can be generated from agar [72]. Agarose-derived oligosaccharides are referred to as agarooligosaccharides (AOSs). There are two forms of AOSs, namely, neo-form and agaro-form. Neo-form AOSs are called neoagarooligosaccharides (NAOSs) and have repeating neoagarobiose units composed of d-galactose at the non-reducing end and 3,6-anhydro-L-galactose (L-AHG) at the reducing end. Table 3 shows the beneficial effects of monosaccharides and oligosaccharides from red algae.

#### 2.3.1. Anti-Melanogenic Activity

Previous studies have reported that NAOSs with different DPs, including neoagarobiose (NeoDP2), neoagarotetraose (NeoDP4) and neoagarohexaose (NeoDP6), had a whitening effect and inhibited TYR activity in murine melanoma B16F10 cells [80,81,82]. NAOSs with different DPs were not cytotoxic to B16F10 up to 100 μg/mL, showing that their skin-whitening effect was not derived from affecting cell viability. In addition, NeoDP4 and NeoDP6 reduced extracellular melanin contents in B16F10 cells and pigmentation evaluated by Fontana-Masson staining in HEMs, whereas agarotriose (DP3), agaropentaose (DP5) and agaroheptaose (DP7) did not reduce melanin production [74].

Recent studies have reported that oligosaccharides from agarose showed anti-melanogenic activity according to the DP of the galactosyl groups [83]. d-glucose and d-galactose are common mono-saccharides of marine algae. L-AHG is major component of agar, while D-AHG is a major monomeric sugar unit of carrageenan from red macroalgae. Previously, effects of monosaccharides including L-AHG, D-AHG and d-galactose on α-MSH-induced melanin production in B16F10 melanoma cells have been reported [74,75]. The melanin level was significantly suppressed by 100 μg/mL of L-AHG. D-AHG also showed an inhibitory effect on melanin production only at 100 μg/mL, but its effect was slightly lower than that of L-AHG. Another monomeric sugar, d-galactose, did not exert any significant reduction in melanin production in B16F10 cells. In addition, a previous study reported that TYR activity was promoted by d-galactose, but it seems likely to be decreased in the presence of glucose [73]. d-glucose also did not affect melanin content in murine melanoma cells [73]. Furthermore, a recent study has demonstrated that L-AHG suppresses melanogenic proteins via inhibiting cyclic adenosine monophosphate/cyclic adenosine monophosphate-dependent protein kinase, MAPK, and Akt signaling pathways in HEMs [84]. Collectively, red macroalgal sugars, such as L-AHG and D-AHG, showed anti-melanogenic activity and are considered to be active components of red macroalgae for skin-whitening activity.

#### 2.3.2. Skin Anti-Inflammatory Activity

An effect of L-AHG on LPS-induced NO production in RAW264.7 cells has been reported [75]. To our knowledge, this was the first report on the biological activity of L-AHG. Nitrite production was significantly suppressed by 100 and 200 μg/mL of L-AHG. D-AHG showed a nitrite-suppressing effect only at 200 μg/mL, but its effect was significantly lower than that of L-AHG. Other saccharides, such as NeoDP2 and d-galactose, did not induce any significant reduction in the nitrite production of RAW264.7 cells.

Enoki et al. reported the anti-inflammatory activities of AOSs including agarobiose (DP2), agarotetraose (DP4) and agarohexaose (DP6), which have L-AHG at the reducing end. Agarobiose (DP2), agarotetraose (DP4) and agarohexaose (DP6) dose-dependently suppressed NO production in RAW264.7 cells [77]. Meanwhile, neo-agarohexaose (DP6), which has d-galactose at the reducing end, had no inhibitory effect on nitrite production. Agarobiose (DP2) suppressed LPS-induced prostaglandin E2 (PGE2), and pro-inflammatory cytokine levels in activated monocytes/macrophages via heme oxygenase-1 (HO-1) induction.

A later study conducted by Enoki et al. demonstrated the anti-inflammatory effects of AOSs mixed with DP 2, 4, 6 and 8 in human monocytes [78]. The AOS mixture attenuated LPS-induced NO levels in human monocytes. Agarobiose (DP2) and agarohexaose (DP6) decreased LPS-induced mRNA levels of COX-2, mPGES-1 in human monocytes. However, it is currently unclear whether AOSs can elicit anti-inflammatory activity in vivo by contacting activated monocytes/machrophages at an inflammation site, since a high dose of AOSs was needed to inhibit the release of pro-inflammatory mediators in an in vitro study.

#### 2.3.3. Antioxidant Activity

Ajisaka et al. compared the antioxidative potency of various carbohydrates including fucoidan and AOSs [85]. In a DPPH assay, fucoidan showed remarkable radical scavenging activity, although lower than ascorbic acid, but AOSs showed almost no DPPH radical scavenging activity up to 20 mM. Notably, the SOD activity assay revealed that AOS had high antioxidant activity, showing almost half of the antioxidant activity of ascorbic acid.

Chen et al. evaluated the antioxidant activity of AOSs with different DPs in cell-based systems [76]. An in vitro DPPH assay revealed that agarohexaose showed the highest radical scavenging capacity. Intracellular ROS levels were investigated using the dichlorofluorescein (DCF) assay in L-02 human liver cells. Agarohexaose at 1 mg/mL significantly reduced H_2_O_2_-induced oxidants up to 50%, showing the highest scavenging capability. In conclusion, AOSs may be novel antioxidants which could protect against cell damage caused by ROS, especially agarohexaose which exhibited excellent effects.

#### 2.3.4. Moisturizing Activity

Previously, NeoDP2 has been reported to show not only whitening effects but also moisturizing effects [80]. NeoDP2 showed a higher hygroscopic ability than glycerol or HA, typical moisturizing reagents, indicating that algae-derived saccharides could be used as a moisturizer in cosmetics.

#### 2.3.5. Anti-Skin Cancer Activity

The ability of AOSs from red macroalgae to prevent tumor promotion in the two-stage mouse skin carcinogenesis model has been reported previously [78]. AOS feeding led to delayed DMBA/TPA-induced tumor incidence and tumor number in Institute of Cancer Research (ICR)mice. PGE2 production was also suppressed by AOS intake in a TPA-induced ear edema model. AOSs downregulated cyclooxygenase-2 (COX-2) and microsomal PGE synthase-1 (mPGES-1), rate-limiting enzymes in PGE2 production, in human monocytes. Consequently, AOSs are expected to prevent tumor promotion by inhibiting PGE2 elevation in chronic inflammation sites.

## 3. Concluding Remarks

In this review, we have presented evidence that various biological activities of marine algae extracts and marine algal carbohydrates act as novel cosmeceuticals. Marine algae extracts and carbohydrates were categorized by source (species), structural parameters, bioactive functions and mechanism. Numerous in vitro and in vivo studies showed that marine algae extracts and algal carbohydrates showed various biological activities against skin disorders including hyperpigmentation, wrinkles, dry skin disorders, skin inflammation and skin cancer. However, although diverse biological activities of marine carbohydrates have been determined, their detailed molecular mechanisms and target proteins are not fully understood. Therefore, further investigations to elicit the precise molecular basis for the biological activity of marine algal compounds should be undertaken. Recently, bioinformatics has been used to screen functional materials derived from natural resources more rapidly and to predict the mechanisms of biological actions [86,87,88]. Thus, using a bioinformatics approach will be a good strategy for finding and understanding more effective marine algal compounds, which will contribute to the development of novel cosmeceuticals.

## Figures and Tables

**Table 1 marinedrugs-16-00459-t001:** Bioactive functions of marine algal extracts.

Species	Solvent	Function	Mechanism	Ref.
*Endarachne binghamiae* *Sargassum siliquastrum* *Ecklonia cava*	A	Anti-melanogenesis	In vitro (B16F10 cells) Mushroom TYR activity (↓) Melanin content (↓) Cellular TYR activity (-)	[16]
*S. siliquastrum* *E. cava*			In vivo (Zebrafish) Melanin content (↓) TYR activity (↓)	
*Ishige okamurae Yendo*	A	Anti-melanogenesis	In vitro (B16F10 cells) Mushroom TYR activity (↓) Melanin content (↓)	[17]
*Sargassum polycystum* *Padina tenuis*	E, H	Anti-melanogenesis	In vitro (HEMs) Mushroom TYR activity (↓) In vivo (Guinea pigs) Melanin content (↓)	[18]
*Schizymenia dubyi*	A	Anti-melanogenesis	In vitro (B16F10 cells) Mushroom TYR activity (↓) Melanin content (↓)	[16]
*Sargassum wightii* *Padina gymnospora*	M, C, EAc, A	Antioxidant	In vitro DPPH radical (↓) Ferrous ion chelation	[19]
*Caulerpa peltata*				
*Gelidiella acerosa*				
*Fucus vesiculosus* (Bladder wrack)	A	Skin anti-aging	In vivo (human cheek skin) Thickness (↑) Elasticity (↑)	[20]
*Blue Lagoon coccoid* *Filamentous*	PBS w/o Mg and Ca (pH 7)	Skin anti-aging Skin barrier function	In vitro (HEKs, HDFs) Gene expression of INV, LOR, TGM-1, FLG (↑) UVA-induced expression of MMP-1 (↓) type 1 collagen (↑)	[21]
			In vivo (Human skin) UVA-induced expression of MMP-1 (↓) type 1 collagen (↑) level of TEWL (↓)	
*Botryococcus braunii*	A	Antioxidant	In vitro (NIH3T3 cells) ORAC (↑), ROS level (↓) DNA damage (↓)	[22]
		Skin anti-aging	In vitro (HaCaT cells) Expression of AQP3, FLG, INV and type 1 and 3 pro-collagen (↑)	
		Anti-inflammation	In vitro (RAW 264.7 cells) iNOS expression (↓) NO production (↓)	
*Chlorella vulgaris*	A	Anti-atopic dermatitis	In vivo (NC/Nga mice) DFE-induced AD (↓) Epidermal thickness (↓) Skin hydration (↑) Infiltration of eosinophil and mast cell (↓) Serum chemokine levels of TARC and MDC (↓) mRNA level of IL-4, IFN-γ (↓)	[23]
*Chlorella sorokiniana* (ROQUETTE *Chlorella* sp.)	Spring water	Anti-skin inflammation	In vivo (hairless Skh-1 mice) TPA-induced skin inflammation (↓) macroscopic score (↓)	[24]
*Chlorella vulgaris*		Anti-skin cancer	In vivo DMBA-induced skin papillomagenesis (↓) Tumor burden (↓) Cumulative number of skin papillomas (↓) Percent incidence of mice bearing skin papillomas (↓)	[25]
*Schizochytrium* (ROQUETTE *Schizochytrium* sp.)	Spring water	Anti-skin inflammation	In vivo (hairless Skh-1 mice) TPA-induced skin inflammation (↓) Macroscopic score (↓)	[26]
*Porphyra yezoensis* (*laver*)	M	UV protection	In vitro (HaCaT cells) Cell viability (↑) Apoptosis (↓) Activation of JNK, ERK (↓)	[27]
*Porphyra umbilicalis* Vitamins, *Ginkgo biloba*	A	UV protection	In vivo (HRS⁄ J-hairless mice) UVA/UVB-induced DNA damage (↓), erythema (↓), level of p53, caspase-3 (↓)	[28]
*Furcellaria lumbricalis* *Fucus vesiculosus*	A	Skin anti-aging	In vitro (HDFs) Expression of type 1 pro-collagen (↑)	[29]
*Spirulina maxima* *Ulva lactuca* *Lola implexa* with other compounds		Skin anti-aging	In vivo (Human skin) Skin hydrating (↑) Skin firming effects (↑)	[30]

**A**: aqueous extract, **AD**: atopic dermatitis, **AQP3**: aquaporin-3, **C**: chloroform extract, **Ca**: calcium, **DFE**: *Dermatophagoides farinae* extract, **DMBA**: 7,12-dimethylbenz [a] anthracene, **DPPH**: 2,2-diphenyl-1-picrylhydrazyl, **E**: ethanol extract, **EAc**: ethyl acetate extract, **ERK**: extracellular signal–regulated kinase, **FLG**: filaggrin, **H**: hexane extract, **HaCaT cells**: immortalized human keratinocytes, **HDFs**: human dermal fibroblasts, **HEKs**: human epidermal keratinocytes, **HEMs**: human epidermal melanocytes, **IFN-****γ**: interferon-gamma, **IL-4**: interleukin-4, **iNOS**: inducible nitric oxide synthase, **INV**: involucrin, **JNK**: c-Jun N-terminal kinase, **LOR**: loricrin, **M**: methanol extract, **MDC**: macrophage-derived chemokine, **Mg**: magnesium, **MMP-1**: matrix metalloproteinase-1, **NIH3T3 cells**: mouse embryo fibroblast cells, **NO**: nitric oxide, **ORAC**: oxygen radical absorbance capacity, **PBS**: phosphate-buffered saline, **TARC**: thymus- and activation-regulated chemokine, **TEWL**: transepidermal water loss, **TGM-1**: transglutaminase-1, **TPA**: 12-O-tetradecanoylphorbol-13-acetate, **TYR**: tyrosinase, **UVA**: ultraviolet A, **UVB**: ultraviolet B, **w/o**: without.

**Table 2 marinedrugs-16-00459-t002:** Bioactive functions of marine algal polysaccharides.

Species	Saccharides	Function	Mechanism	Ref.
	Fucoidan	Anti-melanogenesis	In vitro (Mel-Ab cells) Activation of ERK (↓) Melanin content (↓)	[36]
*Sargassum tenerrimum* *Turbinaria conoides*	Fucoidan	Antioxidant	In vitro DPPH radical (↓) Superoxide radical (↓) High total antioxidant and FRAP ability	[37,38,39,40]
*Costaria costata*	Fucoidan	Skin anti-aging	In vitro (HS68 cells) UVB-induced mRNA and pro-tein expression of MMP-1 (↓) type 1 pro-collagen (↑) Activation of ERK, JNK (↓)	[41,42]
	Fucoidan		In vitro (HaCaT cells) Expression of MMP-1 (↓) type 1 pro-collagen (↑)	[43]
*Mekabu*	Fucoidan		In vivo UVB-induced edema (↓) Thickness of prickle cell layer (↓) MMP-1 activity & expression, IFN-γ (↓)	[44]
*Ascophyllum nodosum*	Fucoidan (16 kDa) by acidic hydrolysis		In vitro (HDFs) IL-1β-induced MMP-9, MMP-3 expression/secretion (↓) TIMP-1 (↑)	[45]
			*Ex vivo* (human skin) Elastic fiber degradation (↓) Leukocyte elastase activity (↓)	
*Laminaria cichorioides*	Fucoidan	Anti-atopic dermatitis	In vivo (Nc/Nga mice) DNCB-induced AD (↓) Clinical severity scores (↓) Scratching counts (↓) Epidermis thickness (↓) Mast cell count (↓) Infiltration of mast cells (↓) Serum histamine (↓) Total IgE (↓)	[46]
			in vitro (Human keratinocytes) AD-associated chemokines TARC, MDC, RANTES (↓)	
	Fucoidan		*Ex vivo* IgE production in PBMC from patients with AD (↓) Immunoglobulin germline transcripts of B cells (↓) IgE-secreting cells count (↓)	[47]
*Saccharina japonica*	Fucoidan	Moisturizing	Higher moisture-absorption and moisture-retention ability than HA	[48]
*Laminaria cichorioides*	Fucoidan (water soluble)	Anti-skin cancer	In vitro (JB6 Cl41 cells) EGF or TPA-induced neoplastic cell transformation (↓) Binding of EGF and EGFR (↓)	[49]
*Saccharina longicruris*	Laminaran	Skin anti-aging	In vivo (Kunming SPF mice) UVA+UVB-induced skin dermal thickness (↓) Hyp content (↑) Serum or mRNA level of MMP-1 (↓), TIMP-1 (↑)	[50]
		Dermal tissue-engineered production	Deposition of matrix (↑)	[51]
*Ulva pertusa*	Ulvans	Antioxidant	In vitro Superoxide (↓) Hydroxyl radicals (↓) Reducing power (↑) Metal chelating ability (↑)	[52]
	Acetylated and benzoylated ulvans			[53]
*Ulva* sp.	Crude ulvans (57 kDa) LMW ulvan (4 kDa)	Skin anti-aging	In vitro (HDFs) Hyaluronan production Collagen release (-)	[54]
*Porphyra* sp.	Porphyran	Antioxidant	In vitro Ferrous ion chelating Reducing power (↑) DPPH radical (↓) Superoxide (↓)	[55]
*Porphyra haitanensis*	Porphyran fraction F1 fraction F2		In vivo (Kumming mice) Antioxidant enzyme activity such as MDA (↓), SOD (↑), GSH-Px (↑) lipid peroxidation (↓) TAOC in different organs (↑)	[56,57]
	Porphyran with different MW		In vitro DPPH radical (↓) Reducing power (↑)	[58]
	LMW Porphyran SD, AD, PD, BD		In vitro DPPH radical (↓) Hydroxyl radicals (↓) Superoxide (↓)	[59]
*Porphyra yezoensis*	Porphyran	Anti-inflammation	In vitro (RAW264.7 cells) LPS-induced NO, iNOS level, NF-κB activation, TNF-α, nuclear translocation of p65, phosphorylation and degradation of IκB-α (↓)	[60,61]
*Porphyridium*	Carrageenan	Anti-melanogenesis	In vivo (Guinea pig) Level of melanosome (↓)	[62]
*Commercial*	*ι*(ІІ)-Carrageenan	Antioxidant Photoprotective	In vitro (HaCaT cells) UVB-induced cell death (↓) DCF-DA: Intracellular ROS (↓) DPPH radical (↓)	[63]
*Eucheuma spinosum* (*Eucheuma denticulatum*)	*ι*(V)-Carrageenan			
*Commercial*	*λ*-Carrageenan			
*Eucheuma cottonii* (*Kappaphycus alvarezii*)	*κ*(ІІІ)-Carrageenan			
*Commercial*	*ι*(ІІ)-Carrageenan			
	κ-COSs (37.7 kDa)	Antioxidant	In vitro Superoxide radical (↓) Hydroxyl radical (↓) DPPH radical (↓) Reducing power (↑)	[64]
	κ-COSs (1.2 kDa) SD (0.8 kDa) LAD (1.2 kDa) HAD (1.4 kDa PD (1.1 kDa)		In vitro Superoxide (↓) Hydroxyl radical (↓) DPPH radical (↓) Reducing power (↑) Iron ion chelation (↑) Total antioxidant activity (↑)	[65,66]
	κ-COSs with CP	Photo-protective	In vitro (HaCaT cells, MEFs) UVB-induced damage (↓)	[67]

**AD**: atopic dermatitis, **BD**: benzoylated derivatives, **COSs**: carrageenan oligosaccharides, **CP**: collagen peptide, **DCF-DA**: 2′,7′-dichlorofluorescin diacetate, **DNCB**: 2,4-dinitrochlorobenzene, **DPPH**: 2,2-diphenyl-1-picrylhydrazyl, **EGF**: epidermal growth factor, **EGFR**: epidermal growth factor receptor, **ERK**: extracellular signal–regulated kinase, **FRAP**: ferric reducing antioxidant power, **GSH-Px**: glutathione peroxidase, **HA**: hyaluronic acid, **HaCaT cells**: immortalized human keratinocytes, **HDFs**: human dermal fibroblasts, **HS68 cells**: human foreskin fibroblast, **Hyp**: hydroxyproline, **IgE**: immunoglobulin E, **I****κB-α**: inhibitor of kappa B, **IL-1β**: interleukin-1β, **IFN-γ**: interferon-gamma, **iNOS**: inducible nitric oxide synthase, **iota 2 [****ι(****ІІ)],****iota 5 [****ι (****V)],****JB6 cells**: mouse epidermal cells, **JNK**: c-Jun N-terminal kinase, **kappa (****κ),****lambda (****λ)****kappa [****κ(****ІІІ****)],****LMW**: low molecular weight, **LPS**: lipopolysaccharide, **MDA**: malondialdehyde, **MDC**: macrophage-derived chemokine, **MEF**: mouse embryonic fibroblasts, **Mel-Ab cells**: immortalized murine melanocyte cell line, **MMP-1**: matrix metalloproteinase-1, **MMP-3**: matrix metalloproteinase-3, **MMP-9**: matrix metalloproteinase-9, **MW**: molecular weight, **NF-κB**: nuclear factor kappa B, **NO**: nitric oxide, **PBMC**: peripheral blood mononuclear cell, **PD**: phosphorylated derivatives, **RANTES**: regulated upon activation, normal T-cell expressed and secreted chemokine, **ROS**: reactive oxygen species, **SD**: sulfated derivatives, **SOD**: superoxide dismutase, **SPF**: specific pathogen free, **TAOC**: total antioxidant capacity, **TARC**: thymus- and activation-regulated chemokine, **TIMP-1**: tissue inhibitor of metalloproteinases inhibitor 1, **TNF**: tumor necrosis factor, **TPA**: 12-O-tetradecanoylphorbol-13-acetate, **UVA**: ultraviolet A, **UVB**: ultraviolet B.

**Table 3 marinedrugs-16-00459-t003:** Bioactive functions of marine algal monosaccharides and oligosaccharides.

DP	Name	Mode of Linkage	Function	Mechanism	Ref.
1	D-Glucose	-	Anti-melanogenesis	In vitro (B16 cells) TYR activity (↓) Melanin content (-)	[73]
	L-AHG	-		In vitro (B16F10 cells or HEMs) Melanin content (↓), TYR activity (-)	[74,75]
	D-AHG	-	Anti-inflammation	In vitro (Raw264.7 cells) LPS-induced NO level (↓)	[75]
	D-Galactose	**-**	Melanogenesis	In vitro (B16 cells) Melanin content (-) TYR activity (↑)	[73,75]
2	Agarobiose	Gal_β__1_→_4_AHG	Antioxidant	In vitro DPPH radical (↓)	[76]
			Anti-inflammation	In vitro (RAW264.7 cell) LPS-induced level of NO, PGE2 (↓) Expression of HO-1 (↑) Protein level of iNOS (↓)	[77]
				In vitro (Human Monocytes) LPS-induced Cytokines TNF-α, IL-1b, IL-6 (↓)	
				In vitro (Human Monocytes) LPS-induced NO level (↓) mRNA level of COX-2, mPGES-1 (↓)	[78]
	Neoagarobiose	AHG_α1_→_3_Gal	Anti-melanogenesis	In vitro (B16 cells) Melanin content (↓) Cellular TYR activity (↓)	[79,80]
			Moisturizing	Higher moisture-absorption and moisture-retention ability than HA	
3	Agarotriose	Gal_β__1_→_4_AHG_α1_→_3_Gal	N.a.	-	-
	Neoagarotriose	AHG_α1_→_3_Gal_β__1_→_4_AHG	N.a.	-	-
4	Agarotetraose	Gal_β__1_→_4_AHG_α1_→_3_Gal_β__1_→_4_AHG	Antioxidant	In vitro DPPH radical (↓)	[76]
			Anti-inflammation	In vitro (RAW264.7 cell) LPS-induced level of NO (↓)	[77]
	Neoagarotetraose	AHG_α1_→_3_Gal_β__1_→_4_AHG_α1_→_3_Gal	Anti-melanogenesis	In vitro (B16 cells or HEMs) Melanin content (↓) Cellular TYR activity (↓)	[74,81]
5	Agaropentaose	Gal_β__1_→_4_AHG_α1_→_3_Gal_β__1_→_4_AHG_α1_→_3_Gal	N.a.	-	-
	Neoagaropentaose	AHG_α1_→_3_Gal_β__1_→_4_AHG_α1_→_3_Gal_β__1_→_4_AHG	N.a.	-	-
6	Agarohexaose	Gal_β__1_→_4_AHG_α1_→_3_Gal_β__1_→_4_AHG_α1_→_3_Gal_β__1_→_4_AHG	Antioxidant	In vitro DPPH radical (↓)	[76]
			Anti-inflammation	In vitro (RAW264.7 cell) LPS-induced level of NO (↓)	[77]
				In vitro (Human Monocytes) LPS-induced NO level (↓) mRNA level of COX-2, mPGES-1 (↓)	[78]
	Neoagarohexaose	AHG_α1_→_3_Gal_β__1_→_4_AHG_α1_→_3_Gal_β__1_→_4_AHG_α1_→_3_Gal	Anti-melanogenesis	In vitro (B16 cells or HEMs) Melanin content (↓) Cellular TYR activity (↓)	[74,81,82]
7	Agaroheptaose	Gal_β__1_→_4_AHG_α1_→_3_Gal_β__1_→_4_AHG_α1_→_3_Gal_β__1_→_4_AHG_α1_→_3_Gal	N.a.	-	-
	Neoagaroheptaose	AHG_α1_→_3_Gal_β__1_→_4_AHG_α1_→_3_Gal_β__1_→_4_AHG_α1_→_3_Gal_β__1_→_4_AHG	N.a.	-	-
8	Agarooctaose	Gal_β__1_→_4_AHG_α1_→_3_Gal_β__1_→_4_AHG_α1_→_3_Gal_β__1_→_4_AHG_α1_→_3_Gal_β__1_→_4_AHG	Antioxidant	In vitro DPPH radical (↓)	[76]
	Neoagarooctaose	AHG_α1_→_3_Gal_β__1_→_4_AHG_α1_→_3_Gal_β__1_→_4_AHG_α1_→_3_Gal_β__1_→_4_AHG_α1_→_3_Gal	N.a.	-	-
9	Agarononaose	Gal_β__1_→_4_AHG_α1_→_3_Gal_β__1_→_4_AHG_α1_→_3_Gal_β__1_→_4_AHG_α1_→_3_Gal_β__1_→_4_AHG_α1_→_3_Gal	N.a.	-	-
	Neoagarononaose	AHG_α1_→_3_Gal_β__1_→_4_AHG_α1_→_3_Gal_β__1_→_4_AHG_α1_→_3_Gal_β__1_→_4_AHG_α1_→_3_Gal_β__1_→_4_AHG	N.a.	-	-
10	Agarodecaose	Gal_β__1_→_4_AHG_α1_→_3_Gal_β__1_→_4_AHG_α1_→_3_Gal_β__1_→_4_AHG_α1_→_3_Gal_β__1_→_4_AHG_α1_→_3_Gal_β__1_→_4_AHG	Antioxidant	In vitro DPPH radical (↓)	[76]
	Neoagarodecaose	AHG_α1_→_3_Gal_β__1_→_4_AHG_α1_→_3_Gal_β__1_→_4_AHG_α1_→_3_Gal_β__1_→_4_AHG_α1_→_3_Gal_β__1_→_4_AHG_α1_→_3_Gal	N.a.	-	-
-	Mixture of AOSs with DP 2, 4, 6 and 8	[Gal_β__1_→_4_AHG]_n_	Anti-melanogenesis	In vitro (B16 cells) Melanin content (↓) Cellular TYR activity (↓)	[82]
			Anti-skin cancer	In vivo (ICR mice) DMBA/TPA-induced tumor incidence (↓), number of papilloma (↓), TPA-induced ear edema (↓) TPA-induced PGE2 (↓)	[78]
			Anti-inflammation	In vitro (Human monocytes) LPS-induced NO level (↓)	

**AOSs**: agaro-oligosaccharides, **B16(F10) cells**: mouse melanoma B16(F10) cells, **COX-2**: cyclooxygenase-2, **D-AHG**: 3,6-anhydro-D-galactose, **DMBA**: 12-dimethylbenz [a] anthracene, **DP**: degree of polymerization, **DPPH**: 2,2-diphenyl-1-picrylhydrazyl, **HA**: hyaluronan, **HEMs**: human epidermal melanocytes, **HO-1**: heme oxygenase-1, **IL**: interleukin, **iNOS**: Iiducible nitric oxide synthase, **L-AHG**: 3,6-anhydro-L-galactose, **LPS**: lipopolysaccharides, **mPGES-1**: microsomal prostaglandin E synthase-1, **N.a.**: not applicable, **NO**: nitric oxide, **PGE2**: prostaglandin E2, **TNF**: tumor necrosis factor, **TPA**: 12-O-tetradecanoylphorbol-13-acetate, **TYR**: tyrosinase, **(-)**: not effective.

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
