# Peer review of "Beneficial Effects of Marine Algae-Derived Carbohydrates for Skin Health"

_marinedrugs, 2018, doi:10.3390/md16110459_

Round 1

Reviewer 1 Report

In this work, the major carbohydrates from marine algae and their properties for various skin beneficial effects were reviewed.

The manuscript is well designed but several changes or mistakes that they must be included or corrected, e.g.:

1) Table 1. Ref. 18, according to line 85, hexane (H ?) could be added as solvent.

2) Table 1: References 30 and 31: Why is included brown, red, blue...?

3) Footnote in Tables: DPPH is 2,2-diphenyl-1-picrylhydrazyl (with a final “l”); TPA : 12-O-tetradecanoylphorbol-13-acetate; DNCB: 2,4-dinitrochlorobenzene; HaCaT cells: Spontaneously (??) immortalized human keratinocytes; Mel-Ab cells: Spontaneously (??) immortalized murine melanocyte cell line.

4) Line 91: Is ethyl acetate (EAc), not E.

5) Line 93: Is it E or EAc???

6) What is PU in line 106?

7) Line 109: Please, to delete T. after Bader.

8) Line 144: Please, to delete S. after Hidalgo-Lucas.

9) Line 148: Reference 26 or 25 ??? And reference 26 in line 150 after tissues ????

10) Please, to include reference 27 in paragraph 153-158.

11) Authors use anti-oxidant and antioxidant. Please, choose one for all manuscript.

12) Table 2: References 42-43: Is it EKR or ERK ??

13) Table 2: Some abbreviations were not included in the footnote, such as: PBMC, HDF, SOD, MDA, TAOC, TNF, NO, iNOS, DCF-DA.... Please, revise all.

14) Table 2: The abbreviation IP is in the footnote, but not in the table.

15) Table 2, reference 5º: it is neoplastic cell (not neopla-stic cell)

16) Lines 194 -196: Kang et al is not the reference 37.

17) Line 209: Is it EKR or ERK ??

18) Table 3: The abbreviation PGES was not included in the footnote.

19) Table 3: L-AHG: 3,6-anhydro-L-galactose, D-AHG: 3,6-anhydro-D-galactose.

20) Line 403: Enoki et al. [78] reported...

21) Line 410: conducted by Enoki et al. [79] demonstrated...

22) Line 418: Please, to delete K. after Ajisaka.

23) Line 423: Please, to delete HV. after Chen.

24) References: All scientific names of algae and the words ‘in vivo’ and ‘in vitro’ in italics.

25) Finally, the abbreviations in the footnote of the Tables must be ordered alphabetically; it is very difficult search the abbreviation and the meaning.

Author Response

Response to Reviewer 1 Comments

Point 1: Table 1. Ref. 18, according to line 85, hexane (H ?) could be added as solvent.

Response 1: Your point is right. The content is a summary of the research of ethanol or hexane extracts, so we added H as a solvent in reference 18 of Table 1 and abbreviation H to the footnotes of Table 1.

Point 2: Table 1: References 30 and 31: Why is included brown, red, blue...?

Response 2: We had originally intended to classify Table 1 into the type according to their pigments. Reference 30 and 31 were studies on mixtures of macroalgae of various species, and were included in the category of mixture instead of the color-dependent category. Therefore, we used the parentheses to mark the type of color separately for reference 30 and 31. However, we have deleted such classification in the submitted manuscript because it is not considered important in the discussion, but we were not able to correct that. Thanks to your point, we deleted that in revised manuscript.

Point 3: Footnote in Tables: DPPH is 2,2-diphenyl-1-picrylhydrazyl (with a final “l”); TPA: 12-O-tetradecanoylphorbol-13-acetate; DNCB: 2,4-dinitrochlorobenzene; HaCaT cells: Spontaneously (??) immortalized human keratinocytes; Mel-Ab cells: Spontaneously (??) immortalized murine melanocyte cell line.

Response 3: We revised the footnotes of Tables as you pointed.

Point 4: Line 91: Is ethyl acetate (EAc), not E.

Point 5: Line 93: Is it E or EAc?

Response 4 and 5: We have previously defined the abbreviation E as ethanol and EAc as ethyl acetate. As suggested, we modified E to EAc at line 100 and 102.

Point 6: What is PU in line 106?

Response 6: We incorrectly labelled the Porphyra umbilicalis mentioned in line 111 as PU. So, the PU on line 115 was changed to P. umbilicalis in revised manuscript.

Point 7: Line 109: Please, to delete T. after Bader.

Response 7: We removed T. from line 165.

Point 8: Line 144: Please, to delete S. after Hidalgo-Lucas.

Response 8: It also deleted S in line 200 as you pointed out.

Point 9: Line 148: Reference 26 or 25 ??? And reference 26 in line 150 after tissues ?

Response 9: According to Reference 25, Chlorella pyrenoidosa reduces the MMP-1 activity, a major marker of skin aging. However, it is just a result of depending only on the measurement of in vitro enzymatic activity. Therefore, we tried to exclude this reference because it was deemed insufficient to demonstrate the anti-skin aging effect of chlorella, but we did not remove it on Table 1. Thanks to your sharp observation, we removed the row of Table 1 and reference 25 in modified manuscript.

Point 10: Please, to include reference 27 in paragraph 153-158.

Response 10: We inserted a citation in the paragraph about reference 26 (line 210).

Point 11: Authors use anti-oxidant and antioxidant. Please, choose one for all manuscript.

Response 11: We also agree with your comment. Therefore, we checked throughout the manuscript and revised anti-oxidant to antioxidant.

Point 12: Table 2: References 42-43: Is it EKR or ERK ??

Response 12: There was a typing error. We corrected the error in Table 2.

Point 13: Table 2: Some abbreviations were not included in the footnote, such as: PBMC, HDF, SOD, MDA, TAOC, TNF, NO, iNOS, DCF-DA.... Please, revise all.

Response 13: Considering your points, we revised the footnotes in the tables.

Point 14: Table 2: The abbreviation IP is in the footnote, but not in the table.

Response 14: Considering your points, we revised the footnotes in the tables.

Point 15: Table 2, reference 5º: it is neoplastic cell (not neopla-stic cell)

Response 15: By your point, we corrected the error in Table 2 (ref. [49]).

Point 16: Lines 194 -196: Kang et al is not the reference 37.

Response 16: This is also our mistake. We modified the author name to match the reference (Line 268).

Point 17: Line 209: Is it EKR or ERK?

Response 17: As with response 12, we revised the typing error in line 283.

Point 18: Table 3: The abbreviation PGES was not included in the footnote.

Response 18: We have already mentioned the full name of PGES in the footnote of Table 3 as follows: mPGES-1: Microsomal prostaglandin E synthase-1.

Point 19: Table 3: L-AHG: 3,6-anhydro-L-galactose, D-AHG: 3,6-anhydro-D-galactose.

Response 19: We revised the error with your kind confirmation.

Point 20: Line 403: Enoki et al. [78] reported...

Point 21: Line 410: conducted by Enoki et al. [79] demonstrated...

Point 22: Line 418: Please, to delete K. after Ajisaka.

Point 23: Line 423: Please, to delete HV. after Chen

Response 20-23: As you pointed out, we modified the author name (line 634, 641, 649 and 654).

Point 24: References: All scientific names of algae and the words ‘in vivo’ and ‘in vitro’ in italics.

Response 24: We did not check the font form in the references. We changed some words into italics thanks to your advice.

Point 25: Finally, the abbreviations in the footnote of the Tables must be ordered alphabetically; it is very difficult search the abbreviation and the meaning.

Response 25: Considering your points, we revised the footnotes in the tables.

Reviewer 2 Report

I recommend that the authors review the English of the article. There are errors in grammar that prevent understanding. Additionally, the references are not formatted according to the requirements. There is an extra comma after the first name or middle name's initial of the last author's, which shouldn't be there, and the volume doesn't appear in Italics.

Some technical terms may need to be revised as well. For example, dehydrated skin disease could be just called dry skin. Or anti-skin aging could be just called anti-aging.

As far as content goes, I believe the manuscript is well organized and logical. There is a considerable overlap between the beneficial biological effects of polysaccharides and mono/oligosacchararides. The authors may want to consider merging section 2.2 and 2.3, having poly- and mono/oligosaccharides in one section. This is just a suggestion.

Author Response

Response to Reviewer 2 Comments

Point 1: I recommend that the authors review the English of the article. There are errors in grammar that prevent understanding. Additionally, the references are not formatted according to the requirements. There is an extra comma after the first name or middle name's initial of the last author's, which shouldn't be there, and the volume doesn't appear in Italics.

Response 1: Our manuscript was checked by a native English speaking colleague, and the changes were marked using "Track Changes" in Microsoft Word. Also, as you suggested, we checked the format of the references and revised the error by editing the EndNote style.

Point 2: Some technical terms may need to be revised as well. For example, dehydrated skin disease could be just called dry skin. Or anti-skin aging could be just called anti-aging.

Response 2: As suggest, we revised 'dehydrated skin' to 'dry skin' in line 680 but didn't change 'anti-skin aging'. Aging involves a variety of mechanisms and symptoms throughout our bodies. However, we focused on specific aging in skin cell or tissues in this review. Therefore, it is more accurate to refer to anti-skin aging.

Point 3: As far as content goes, I believe the manuscript is well organized and logical. There is a considerable overlap between the beneficial biological effects of polysaccharides and mono/oligosacchararides. The authors may want to consider merging section 2.2 and 2.3, having poly- and mono/oligosaccharides in one section. This is just a suggestion.

Response 3: We partially agree with your suggestion. Although the biological effects of the polysaccharides and mono/oligosaccharides from marine algae may overlap, we have divided them into two sections. Since most of the polysaccharides in marine algae are nondigestible and nondegradable, we expected that the polysaccharides will work as itself rather than oligosaccharide or monosaccharide by metabolism. Thus, their effects on regulation of biological activities can be the results of different actions, so we discussed the effects in separating sections.

Reviewer 3 Report

This review article is an admirable attempt to collect the most important information about the potential novel applications of marine algae-derived carbohydrates as bioactive constituents for cosmetic and cosmeceutical products. Analysis about the compounds responsible of their activity on skin are also analysed. The subject is of current interest and therefore is appropriate to have the authors' point of view and revision on this issue because they have knowledge and expertise in this field of study. The manuscript fits within the scope of the journal. Nevertheless, this reviewer has some suggestions that could help to improve the work and to wake up the interest of a wider scientific community on this work:

1. Any criticism on problems derived from the use of bioactive ingredients from natural sources is completely lacking. This is not necessarily a part of the manuscript but neither is an uncritical list of potential benefits.

2. Although bioactive ingredients from marine algae are substances with a general low risk and favorable perception by society as compared with synthetic substances, a risk assessment of their hazard is always necessary.

3. Concerning the possibility to use some compounds from marine algae as sunscreen, explore this issue.

4. The safety of pharmaceutic, cosmetic and other chemical products can be assessed using literature data, in vitro studies and human tests. However, computational studies can also provide some insight into the problem. Moreover, some of the problems arising from the arrival to the environment of these components can be addressed by computational methodologies. Relevant reviews focused on these computational approaches have treated these concerns (e.g. "J. Agric. Food Chem. 2017, 65, 2017-2018"; "Molecules 2016, 21(6), 748"; etc.). Currently, this information is scarce and further valuable information to the readers is required in order to wake up the interest of a wider scientific community and offer a whole vision of the importance of this issue.

5. The marine algae go through different transformation processes. These processes can originate waste. This reviewer misses some mention to the occurrence of further degradates from, for example, the treatment of wastewater. In this context, more discussion focused on degradation products and legislative requirements about the effluents would be appreciate.

Author Response

Response to Reviewer 3 Comments

Point 1: Any criticism on problems derived from the use of bioactive ingredients from natural sources is completely lacking. This is not necessarily a part of the manuscript but neither is an uncritical list of potential benefits.

Response 1: Your point is valid and it raises good concerns. However, we sought to focus on the carbohydrates from marine algae among many studies about marine natural sources for development of functional ingredients and to provide the insights into their beneficial effects on skin health. We regard that our discussions in this review meet these objectives. We did not consider the sunscreen section because we sought to discuss biological activity for skin health based on cellular biological regulation in the review.

Point 2: Although bioactive ingredients from marine algae are substances with a general low risk and favorable perception by society as compared with synthetic substances, a risk assessment of their hazard is always necessary.

Response 2: We also suggest that safety should be determined prior to functionality in the study of bioactive ingredients. Although we have mentioned the safety for some of the carbohydrates discussed in this review, marine algae is a food material that is easily accessible to the diet, and we can predict that the hazard of polysaccharides, its major component, will be minimal.

Point 3: Concerning the possibility to use some compounds from marine algae as sunscreen, explore this issue.

Response 3: Sunscreen has a role to block the UV to protect the skin, and acts through a mechanisms that reflects and scatter UV, or absorbs and emits UV to other energy. The topic in your comment is interesting subject because many unhealthy skin symptoms such as inflammation, aging and pigmentation can be induced by UV. However, the mechanism of sunscreen approaches from chemicophysical mechanisms rather than cellular biology based mechanisms. Instead, we reviewed the photo-protective effects, including antioxidant activity and suppression of UV-induced cell damage, as a function of marine algae-derived carbohydrates.

Point 4: The safety of pharmaceutic, cosmetic and other chemical products can be assessed using literature data, in vitro studies and human tests. However, computational studies can also provide some insight into the problem. Moreover, some of the problems arising from the arrival to the environment of these components can be addressed by computational methodologies. Relevant reviews focused on these computational approaches have treated these concerns (e.g. "J. Agric. Food Chem. 2017, 65, 2017-2018"; "Molecules 2016, 21(6), 748"; etc.). Currently, this information is scarce and further valuable information to the readers is required in order to wake up the interest of a wider scientific community and offer a whole vision of the importance of this issue.

Response 4: A recent computational approach has been added to the conclusion section in the manuscript, and related references have been added.

Point 5: The marine algae go through different transformation processes. These processes can originate waste. This reviewer misses some mention to the occurrence of further degradates from, for example, the treatment of wastewater. In this context, more discussion focused on degradation products and legislative requirements about the effluents would be appreciate.

Response 5: Thank you for your comment. However, in this review, we focused on the biological activity of marine algal carbohydrates for skin health based on cellular biological regulation. Thus, the reviewers point is somewhat different from the subject of our review paper.